# Computational Modeling of Boundary Layer Flashback in a Swirling Stratified Flame Using a LES-Based Non-Adiabatic Tabulated Chemistry Approach

**DOI:** 10.3390/e23050567

**Published:** 2021-05-02

**Authors:** Xudong Jiang, Yihao Tang, Zhaohui Liu, Venkat Raman

**Affiliations:** 1Department of Aerospace Engineering, University of Michigan, Ann Arbor, MI 48109, USA; yihao.tang@ansys.com (Y.T.); ramanvr@umich.edu (V.R.); 2U.S.-China Clean Energy Research Center, University State Key Laboratory of Coal Combustion, School of Energy and Power Engineering, Huazhong University of Science and Technology, Wuhan 430074, China; zliu@hust.edu.cn

**Keywords:** turbulent combustion, flashback, tabulated chemistry, heat loss

## Abstract

When operating under lean fuel–air conditions, flame flashback is an operational safety issue in stationary gas turbines. In particular, with the increased use of hydrogen, the propagation of the flame through the boundary layers into the mixing section becomes feasible. Typically, these mixing regions are not designed to hold a high-temperature flame and can lead to catastrophic failure of the gas turbine. Flame flashback along the boundary layers is a competition between chemical reactions in a turbulent flow, where fuel and air are incompletely mixed, and heat loss to the wall that promotes flame quenching. The focus of this work is to develop a comprehensive simulation approach to model boundary layer flashback, accounting for fuel–air stratification and wall heat loss. A large eddy simulation (LES) based framework is used, along with a tabulation-based combustion model. Different approaches to tabulation and the effect of wall heat loss are studied. An experimental flashback configuration is used to understand the predictive accuracy of the models. It is shown that diffusion-flame-based tabulation methods are better suited due to the flashback occurring in relatively low-strain and lean fuel–air mixtures. Further, the flashback is promoted by the formation of features such as flame tongues, which induce negative velocity separated boundary layer flow that promotes upstream flame motion. The wall heat loss alters the strength of these separated flows, which in turn affects the flashback propensity. Comparisons with experimental data for both non-reacting cases that quantify fuel–air mixing and reacting flashback cases are used to demonstrate predictive accuracy.

## 1. Introduction

Stationary gas turbines, driven by the need for reduced emissions of oxides of nitrogen (collectively called NO${}_{x}$), utilize a globally lean and premixed combustion mode. Since NO${}_{x}$ is highly sensitive to temperature, the lean operation reduces the operating temperature, thereby reducing emissions formation. However, this combustion mode is subject to operational stability issues, such as lean blowout [1] or flashback [2]. In order to further reduce greenhouse gas emissions, such lean premixed combustors are operated with hydrogen-containing fuel mixtures, with the focus on high hydrogen content [3]. Due to the broad flammability limits and high reactivity of hydrogen, the aforementioned stability issues can be further exacerbated. In particular, flame flashback is an important operational and safety issue.

Even with hydrocarbon fuels, flashback is one of the most frequently encountered problems in the design of stationary gas turbines [1,4,5]. During flashback, the flame propagates into the premixing zone from the combustion section of the gas turbine and causes damage to the premixing component of the combustor, as the premixed nozzles are not designed to withstand such heat load caused by flame [1]. Flashback may occur through four different modes [2]: (a) through the core flow, where the flow velocity is lower than the burning rate of the fuel–air mixture; (b) through pressure-heat release coupling leading to flame propagation into fresh gases; (c) combustion induced vortex breakdown (CIVB), where the flame propagates through low-velocity regions generated in swirl-stabilized combustors; and (d) boundary layer (BL) flashback, where the flame propagates close to the wall in the low-momentum near-wall flow. The focus of this work is on this last mode of flashback.

In the past, there has been extensive research on boundary layer flashback in a non-swirling channel and pipe flows [6]. Among the earliest efforts, Lewis and Von Elbe [7] correlated temperature distribution and flow pattern to model the flashback limit. This work resulted in the classical critical-gradient model to evaluate propensity for boundary layer flashback, shown schematically in Figure 1. Here, a laminar boundary layer with the velocity (*u*) that is dependent on distance from the wall is assumed. At the same time, due to near-wall flame quenching through conduct heat loss, the flame speed $S_{f}$ is also a function of distance from the wall. $\delta_{p}$ represents the penetration distance, defined as the location where the burning velocity is equal to the flow velocity [6]. As shown in Figure 1, flashback is possible when the local flame velocity is higher than the gas velocity in the boundary layer, which is quantified by the velocity gradient at the wall. In other words, for a given $S_{f}$ profile, there exists a critical velocity gradient below which flashback is theoretically possible. However, the critical gradient model does not consider the effect of the flame on the flow.

More recent studies have shown that the geometry of the flow, as well as the interaction between the flame front and the boundary layer, are key factors in determining flashback propensity. For instance, the flashback process is different in confined and unconfined flows. In unconfined flashback, several studies [7,8] have shown that the flame is stabilized at the burner rim before the onset of flashback and the flame speed is the key driver for the initiation of flashback. A model for unconfined flame flashback proposed by Hoferichter et al. [9] shows that the key process to capture flashback is turbulence, which determines the flame stretch and the burning velocity. In confined flames, the boundary layer separation induced by a pressure rise upstream at the flame tip is the main parameter for the onset of flashback [10], and the classical critical-gradient model becomes less applicable as the boundary layer velocity profile can be strongly disturbed by the adverse pressure gradient induced by the flame front [11]. Eichler et al. [12] performed a systematic study that compared the flashback propensity in unconfined and confined flashback (shown schematically in Figure 2). In general, the flashback propensity has been found to be much higher in the confined configuration compared to unconfined flows [2,12], where the difference is mainly due to the combined effect of heat loss and the pressure gradient. Specifically, the leakage of unburnt gases through the gap between the wall and the flame front reduces flashback propensity by relaxing the pressure rise and enhancing the conductive heat loss in the flame preheat zone. This latter effect is much more prominent in an unconfined flashback than in a confined flashback [2,12]. Other studies show that the flashback near critical condition was found to be insensitive to the thermal expansion (numerical heat release) while sensitive to the wall heat loss [13].

Most gas turbine combustors introduce the fuel–air mixture using swirling inflows, which may also have an impact on flashback. An experimental study in a swirling, bluff-body stabilized, and unconfined flame conducted by Heeger et al. [14] provides insights into the flame upstream propagation in swirling flame. Consistent with the non-swirling experiment, the reduction in density along the flame front generates an adverse pressure gradient, which produces negative velocity beneath the leading tip of the flame, resulting in an upstream propagation of the flame front. However, it is also found that flame can creep into the mixing tube without the negative velocity zone, which implies that the driving mechanism of flashback in a swirling flow is different from that in a non-swirling confined flow. To further probe the dominant mechanism in swirling flashback, Karimi et al. [15] investigated the influence of swirl number and center body radius on flashback using an unconfined, swirling, and bluff-body stabilized premixed flame. It was found that both the swirl number and the center body radius have intensifying effects on flame propagation speed. Furthermore, flashback is found to always be associated with static pressure rise behind the flame front. In a later work [16], it was concluded that the static pressure rise provides a circumferential motion in addition to the axial motion for flashback. Further, Ebi et al. [17] conducted an experimental study focusing on dominant mechanisms for confined swirl-flame boundary layer flashback. They proposed the two concepts of a large-scale “flame tongue” and a small-scale “flame bulge” to describe swirl-flame propagation in a tube with a central body. The flame tongue denotes the large leading part of the propagating flame, where the negative axial velocity area around the flame front mainly concentrates in the radial direction and largely beyond the quenching distance [18]. The flame bulges feature a physical length scale that is an order-of-magnitude smaller than the flame tongue and are scattered on the trailing part of the swirl flame, where the induced reverse flow pocket is radially narrow, but reaches farther upstream compared to the flame tongue. As observed in their experiment, the small-scale flame bulge produces a reverse flow pocket upstream of flame tip through static pressure rise, similar to a non-swirling BL flashback, but it only resists the approaching flow momentarily, being swept downstream by fast approaching flow from the other side. So, the flame bulges were not the drivers of sustained flame upstream propagation around the center body. On the other hand, the flame tongue deflects the approaching flow and generates a negative axial velocity zone in the vicinity of the flame front, which provides axial and azimuthal motion of the flashback flame front. These two concepts are adopted by the current study that focuses on the same flow configuration, and will be discussed further below.

The mixing of fuel and air streams occurs ahead of the burner exit in practical designs. In this mixing region, a homogeneous mixture is not found and variations in equivalence ratio are to be expected. Since the flashback process allows the flame to enter this premixing region, this flame propagation occurs not in fully premixed, but in partially-premixed or stratified mixtures. While fuel stratification has been found to have an influence on the safe operability limits of flashback [19], only limited studies have been conducted. Ranjan et al. [20] investigated the effects of stratification on flashback in the same facility studied by Ebi et al. [17]. They also found that the flashback around the center body is driven by the flame tongue. Specifically, the flashback process is similar to the lean swirl BL flashback in a fully premixed condition, but the flame propagation can be temporarily halted. This feature occurs due to the turbulence and equivalence ratio variation that makes the flame front highly wrinkled. In a later study [21], the effect of hydrogen enrichment on flashback in atmospheric pressure stratified swirl flames was revealed. Different from their previous study [20], the flame front propagated along the outer wall, but was arrested due to the thin flame structures produced by the interaction of the flame tongue and outer wall. This configuration has also been studied experimentally at an elevated pressure of 3 bar by keeping the same inflow velocity (i.e., increased density and Reynolds number) [20,22]. Those results show a similar global behavior of flashback, even though a more wrinkled flame surface is observed due to increased turbulence. The fuel distribution is also found the be different, more concentrated near the central body, and less well-mixed, possibly due to the differences in the turbulence structure.

Due to the inherent limitations in probing such complex flows experimentally, computational methods are needed to provide a complementary approach. Many numerical models have been proposed [23,24,25]. However, the use of numerical simulations for modeling flashback is still limited by two aspects. First, flashback is a transient process with multi-physics interactions including turbulent near-wall flow [23], the interaction between reaction and near-wall quenching [12], and propagation in boundary layers [14,16,17,18]. Second, the geometry of practical gas turbine combustors that employ swirling flows is complex, which introduces additional physical processes. Gruber et al. [23] conducted direct numerical simulations (DNS) in a fully developed turbulent channel premixed hydrogen–air flames to study the flame–wall interaction during flame propagation. It was found that the boundary layer flashback in non-swirling channel flow is associated with boundary separation induced by low-momentum streaks, and the near-wall resolution is necessary for the numerical study to describe near-wall reverse flow and capture flashback. Lietz et al. [24] applied large-eddy simulations (LES) on the same channel configuration to analyze the flame–wall interactions during flashback. They also explored the influence of different sub-grid-scale closure of the turbulent combustion models (i.e., direct flamelet model, algebraic flame surface density model, and filtered tabulated chemistry for LES), as well as the filter width on flashback predictions, and revealed that an appropriate resolution of near-wall turbulence and blockage effect induced by combustion is necessary to capture flashback. In a confined hydrogen–air non-swirling flow, Endres et al. [25] used LES combined with detailed chemical kinetics and heat loss to investigate flame propagation. They accurately reproduced confined flashback limits reported in experiments [12]. Their study also revealed that flashback occurs where the size of the flow separation zone is significantly larger than the quenching distance, which was also proposed by Eichler et al. [2]. In a follow-up study, Endres et al. [26] used the same numerical method and configuration to study the effect of pressure on flashback. With an increase in pressure, the size of the separation zone ahead of the propagating flame front decreases, which leads to a reduction in flashback propensity. Meanwhile, the quenching distance decreases, which results in an increase in flashback propensity. It was further concluded that one-dimensional analyses were not suitable in high-pressure conditions since the pressure rise near the flame front is overestimated due to one-dimensional pressure approximations, while the degree of boundary layer separation is underestimated based on the uniform wall-normal pressure assumption.

From a practical perspective, simulations of flashback in realistic gas turbine combustors are relatively sparse. Lietz [27] carried out LES with filtered tabulated chemistry model on lean premixed flashback in swirling combustor [28]. They concluded that the flamelet-based method can capture some qualitative and statistical properties of the flame propagation into the mixing tube. However, the heat loss between propagating flame and wall was not included, although this process is known to play a dominant role [2,13,25,26].

With this background, the focus of this work is in developing a comprehensive LES modeling approach that accounts for wall heat loss, stratification of the fuel–air mixture in the premixing tube, and the swirling flow typical of modern gas turbines. Since fuel stratification could lead to partially-premixed flame behavior in the mixing tube, even when the downstream combustor operates in premixed mode, different flame structures have to be considered. Therefore, the non-adiabatic flamelet progress variable (FPV) model and the flamelet generated manifold (FGM) are used and compared with experimental data. Further, the effect of heat loss on flashback is systematically studied. Flashback is investigated using various wall temperature conditions that correspond to varying degrees of heat loss. Comparing simulations with different heat loss profiles allows a detailed analysis of the influence of quenching distance and distribution of negative axial velocity on the flashback process. Essentially, the models proposed here are extensions of the FPV and FGM methods, and the baseline performance is similar between the two approaches for canonical flows. Both methods are computationally efficient while including detailed kinetics. The proposed modification maintains a concise set of tabulation parameters, and has the potential to be extended to include more complex phenomena. For instance, with a different boundary condition implementation, conjugate heat transfer to the wall could be captured.

The paper proceeds as follows. The simulation details, including the combustion modeling procedures, are provided next (Section 2), followed by a description of the configuration and computational domain (Section 2.3). The model swirling flow configuration with flashback, which was studied experimentally by Clemens and co-workers [20,28], is used to validate the simulations, and these results are presented in Section 3. Conclusions are provided in Section 4.

## 2. Simulation Details

In this work, the large eddy simulation (LES) approach is used to model turbulent flame propagation. It is now well-established that LES can capture transient and unsteady effects in turbulent combustion [29,30]. However, LES is not particularly suited for modeling near-wall flows, where the local anisotropy of turbulence renders sub-filter models inaccurate. In the studies conducted here, the Reynolds number is sufficiently low, such that the near-wall flow can be adequately resolved by the computational grids used. As a result, conventional isotropic sub-filter models are used away from the wall. The focus here is on the modeling of combustion processes, including heat transfer to the wall that will affect flame stability and the flow field. Below, the tabulation-based combustion modeling approach is first described, followed by the governing equations for fluid flow. Two different models, namely the non-adiabatic flamelet progress variable (FPV) [31] model and the flamelet generated manifold (FGM) [32] model, are used to model turbulent combustion. Two canonical configurations featuring a premixed and a non-premixed laminar flame are considered for constructing the tabulation. Further details are provided below.

### 2.1. FPV Tabulation

The FPV approach includes detailed chemistry via a tabulation technique that is computationally efficient. Essentially, the FPV tabulation approximates the combustion variables of the turbulent flame structure by 1-d flamelet solutions (i.e., the flamelet assumption [33]), which are described as a function of the look-up table controlling parameters, and tracks the evolution of these few parameters instead of the full reaction system. In the original FPV approach [31], the steady counterflow diffusion flamelet formulation was applied to construct the look-up table, where the flamelet solutions Ψ are initially obtained in the physical space as Ψ(x;L), with *x* being the physical coordinate, and *L* is a parameter (the domain length) that affects the strain rate. This solution is then mapped to the tabulation space as Ψ(Z,C), with *Z* being mixture fraction and *C* being a progress variable defined through a suitably chosen set of mass fractions as $C=Y_{H_{2}O}+Y_{CO_{2}}$.

One of the objectives of this work is to determine the influence of tabulation methods on the flashback process. For this purpose, two different canonical flame configurations are considered. In the first approach, one-dimensional counterflow diffusion flames are used to build the tabulation. This conventional FPV-approach-based table is built from a series of 1D calculations of diffusion flames with varying scalar dissipation rates. In the second approach, 1D freely propagating premixed flames are used, with each solution corresponding to a particular equivalence ratio (or mixture fraction). The resulting solution is again mapped into the {Z,C} space. This latter approach is similar to the flamelet generated manifold (FGM) technique [32]. A 28-species detailed chemical kinetic mechanism [34] is applied to perform the flamelet calculations, solved using the FlameMaster package [35]. Figure 3 shows the solutions from the two sets of calculations plotted in the tabulation space. For the purpose of discussion, the two tabulations are referred to as diffusion (obtained from FPV procedure) and premixed (obtained from FGM procedure) databases. The reason for considering these two canonical flames is due to the nature of the flow in the geometry considered here (to be discussed in Section 2.3). Since the fuel is injected using discrete holes, the flame encounters an incompletely-mixed fluid as it moves upstream into the mixing section. Depending on the mixing characteristics and operating conditions, the flame structure can replicate partially-premixed or stratified combustion [36]. While there are alternative modeling techniques for handling such multi-regime combustions [37,38], we have used a simpler description in order to understand the role of the flame modeling on the flashback process. At realistic engine conditions, the boundary layers may be thinner, which will require additional modeling. The effect of chemical kinetics, on the other hand, can be fully included, since detailed kinetics are used to construct the tabulation.

One of the main factors affecting flashback is the heat transfer to the wall. For this reason, the above models have to be modified to include the effect of such heat loss. Many previous studies [39,40,41] have studied non-adiabatic flamelet modeling and have found that performance is usually insensitive to the specific strategy used to introduce heat loss. In this study, heat loss effects are introduced into the flamelet solutions through a sink term added to the flamelet energy equation, which takes the form of conductive heat transfer:(1)$$\dot{q}=-\frac{\lambda }{\delta^{2}}\left(T-T_{wall}\right),$$
where $\dot{q}$ is the energy sink term added to the premixed/diffusion flamelet energy Equation [33,42] and is dependent on the local value of flamelet temperature *T* and thermal conductivity λ. Here, δ and $T_{wall}$ are, respectively, the prescribed length and temperature that are fixed at constant values within each flamelet calculation. This heat loss modeling essentially assumes that the 1d flamelet experiences conductive heat loss to a fictional wall maintaining a constant temperature of $T_{w}$ at a distance of δ. When generating the flamelet solutions, the distance parameter is varied from +∞ (adiabatic) to a sufficiently low value (maximum heat loss) such that the populated flamelet is fully quenched, whereas the wall temperature parameter is set to room temperature. The FPV tabulation strategy introduced above is then performed with this additional sink term and is repeated for different levels of the prescribed δ. As a result, the original 2d look-up table is extended into the 3d phase space of {H,Z,C}, with *H* being enthalpy defect [43]. The enthalpy defect measures the local heat loss value, which is defined as
(2)$$H=h-h_{ad},$$
where *h* is the total enthalpy and $h_{ad}$ is the total enthalpy of an adiabatic flamelet at identical *C* and *Z* values. The advantage of tracking enthalpy defect instead of total enthalpy is that it provides a more straightforward understanding of the spatial distribution of heat loss in a configuration that is fuel-stratified (Section 2.3). Enthalpy itself is a strong function of the local equivalence ratio and will mask the impact of heat loss.

The final tabulations of the non-adiabatic FPV models are shown in Figure 4 and Figure 5 for the diffusion database and premixed database, respectively. For the diffusion database, note that, under sufficient heat loss effects, the scalar dissipation rate response locus is no longer a typical S-shaped curve but instead an O-shaped closed curve [35]. At low scalar dissipation rates, the impact of heat loss increases significantly with the local flow time scale. This reduces the maximum temperature value at the upper stable combustion branch, and causes it to become comparable to that in the lower unstable combustion branch (dash line in Figure 4). This leads to a multi-value issue when mapping the flamelet database onto the tabulation phase space, where the flamelet phase space trajectories cross paths between those obtained from the left half part of the O-shaped locus with that of the right half of the locus. To avoid this issue, in this study, only the right half of the locus is applied to construct the tabulation when an O-shaped locus is encountered. The premixed database is more straightforward, with each set of premixed flamelet solutions computed using a fixed equivalence ratio. Here, δ is varied to compute different solutions, and decreased until a reacting flamelet solution can no longer be obtained (Figure 5), and the process is repeated for multiple sets of flamelet solutions that span equivalence ratios within the lean and rich flammability limits to cover the effective area of the tabulation phase space (Figure 5).

### 2.2. LES Governing Equations

In the LES approach, the flow field is decomposed into resolved and unresolved fields using a spatial filtering operation. Governing equations are solved for the resolved variables, while the effect of the unresolved scales is modeled using statistical closures [29,30,44]. The filtered equation for mass conservation is given by:(3)$$\frac{\partial \bar{\rho }}{\partial t}+\frac{\partial \bar{\rho }{\tilde{u}}_{j}}{\partial x_{j}}=0,$$
where $\bar{\cdot }$ denotes filtered variable, while $\tilde{\cdot }$ denotes Favre or density-weighted filtered variable, ρ is the density, and $u_{j}$ denotes the velocity in the *j*-th direction. The governing equation for filtered momentum is given by
(4)$$\frac{\partial \bar{\rho }{\tilde{u}}_{i}}{\partial t}+\frac{\partial \bar{\rho }{\tilde{u}}_{i}{\tilde{u}}_{j}}{\partial x_{j}}=-\frac{\partial \tilde{P}}{\partial x_{i}}+\frac{\partial \tau_{ij}}{\partial x_{j}}+\frac{\partial T_{ij}}{\partial x_{j}},$$
where *P* is pressure, $\tau_{ij}$ is the viscous stress tensor given by
(5)$$\tau_{ij}=\tilde{\mu }\left(\frac{\partial {\tilde{u}}_{i}}{\partial x_{j}}+\frac{\partial {\tilde{u}}_{j}}{\partial x_{i}}-\frac{2}{3}\frac{\partial {\tilde{u}}_{k}}{\partial x_{k}}\delta_{ij}\right)=2\tilde{\mu }{\tilde{S}}_{ij},$$
and $T_{ij}=\bar{\rho }{\tilde{u}}_{i}{\tilde{u}}_{j}-\bar{\rho }\hat{u_{i}u_{j}}$ is the turbulent stress tensor, which is closed with gradient diffusion hypothesis with eddy viscosity estimated by the dynamic Smagorinsky model [44].

Apart from the mass and momentum equations, three transported scalars for filtered progress variable *C*, mixture fraction *Z*, and enthalpy defect *H* are solved. The transport equations of the filtered scalars of *Z* and *C* can be written as
(6)$$\frac{\partial \bar{\rho }\tilde{\Psi }}{\partial t}+\frac{\partial \bar{\rho }{\tilde{u}}_{j}\tilde{\Psi }}{\partial x_{j}}=\frac{\partial }{\partial x_{j}}\left({\tilde{D}}_{\Psi }\frac{\partial \tilde{\Psi }}{\partial x_{j}}\right)+\frac{\partial \bar{\rho }\left(\tilde{\Psi }{\tilde{u}}_{j}-\tilde{\Psi u_{j}}\right)}{\partial x_{j}}+\bar{\rho }{\tilde{\dot{\omega }}}_{\Psi },$$
where Ψ denotes the transported scalar of {Z,C}, $D_{\Psi }$ is the molecular diffusivity, and ${\dot{\omega }}_{\Psi }$ is the reaction source term that is zero for *Z*, but obtained from look-up table for *C*. The second term on the right-hand side represents turbulent transport, which is closed using the dynamic viscosity model and a constant turbulent Schmidt number of $Sc_{T}=0.72$ for both *Z* and *C*.

The transport equation of enthalpy defect can be derived by subtracting the the adiabatic enthalpy transport equation from the non-adiabatic equation, where the thermal properties in the adiabatic case are denoted by an additional subscript “ad” and the flow properties (e.g., velocity and turbulent viscosity) are assumed to be the same between the adiabatic and the non-adiabatic versions. This derivation then leads to the following Equation [45].
(7)$$\frac{\partial \bar{\rho }\tilde{H}}{\partial t}+\frac{\partial \bar{\rho }{\tilde{u}}_{j}\tilde{H}}{\partial x_{j}}=\frac{\partial }{\partial x_{j}}\left(\tilde{\lambda_{ad}}\frac{\partial \tilde{T_{ad}}}{\partial x_{j}}-\tilde{\lambda }\frac{\partial \tilde{T}}{\partial x_{j}}\right)+\frac{\partial }{\partial x_{j}}\left(\frac{\mu_{T}}{Pr_{T}}\frac{\partial \tilde{H}}{\partial x_{j}}\right),$$
where the adiabatic properties are obtained from FPV/FGM tables with zero enthalpy defect, whereas the non-adiabatic properties are obtained from the same table but by using the local *H*-value. A Neumann boundary condition is assigned to $T_{ad}$. A spatially-varying Dirichlet boundary condition is applied to *T*, where the local boundary face value is set to room temperature when the flame front has not reached its position (defined by C<0.01) and toggled to a preset wall temperature $T_{w}$ once the flame front reaches that position. Note that this treatment is not equivalent to the modeling of conjugate heat transfer: the wall temperature $T_{w}$ does not refer to the actual temperature of the wall, but is a measure of the heat being taken away by the wall. Stated differently, this treatment assumes that the flame front heats the facility wall infinitely fast to the prescribed temperature of $T_{w}$. A series of test cases with different set-ups of $T_{w}$ are performed (Table 2). A constant turbulent Prandtl number of $Pr_{T}=0.72$ is applied.

The flamelet databases are used to obtain the filtered thermal transport coefficients and the reaction source terms. Since only the filtered version of these terms are needed, they are obtained from the tabulation using a convolution operation based on a presumed-PDF approach. Essentially, the sub-filter joint distribution of the scalars (C,Z,H) is assumed to be the product of a β-function (mixture fraction) and delta function (C,H) [43,45]. The mixture fraction PDF is described using the filtered mixture fraction and sub-filter variance. Filtered terms are obtained using the convolution operation:(8)$$\tilde{\phi }\left(\tilde{Z},\tilde{Z^{\prime \prime 2}},\tilde{C},\tilde{H}\right)=\;\int \int \int \;\beta \left(Z;\tilde{Z},\tilde{Z^{\prime \prime 2}}\right)\delta \left(C-\tilde{C}\right)\delta \left(H-\tilde{H}\right)\;dZ\;dC\;dH,$$
where the filtered mixture fraction variance $\tilde{Z^{\prime \prime 2}}$ is modeled by assuming local equilibrium between the turbulence production and dissipation rate, as
(9)$$\tilde{Z^{\prime \prime 2}}=C_{v}\Delta^{2}{\left|\nabla \tilde{Z}\right|}^{2},$$
where Δ is the width and the constant $C_{v}$ is assumed to be 0.1 [46].

The above set of equations are implemented in a low-Mach representation of the governing equations [45,47]. The implementation minimizes turbulent energy dissipation and has been shown to be second-order accurate in time and space [47]. The solver is parallelized using MPI-based domain decomposition.

### 2.3. Configuration and Computational Domain

In this study, the experimental configuration of Rakesh et al. [17,21] is considered. Figure 6 shows a schematic of the model swirl combustor, with a premixing section (mixing tube) and a combustor. For the results presented in this manuscript, a specialized cylindrical coordinate system is applied (Figure 6), with the radial axis origin r=0 set to the inner wall of the mixing tube. Fuel is injected through a set of small ports located on the swirler outer surface, whereas the air stream is introduced from the bottom face of the geometry. The swirler leads enhance mixing of fuel and air in the mixing tube, and also affects the development of the boundary layers on the inner and outer walls. The mixing tube contains a central body, with its length (measured from the swirler trailing edge to the combustor tube entrance), inner diameter, and outer diameter of the mixing tube being 117.5 mm, 25.4 mm, and 52 mm, respectively. Since the fuel ports are located at a radial position biased towards the outer wall, the mixture is fuel-lean near the center body and fuel-rich near the outer wall. The combustion tube has a diameter of 100 mm and a length of 150 mm. The cases studied here were conducted at atmospheric pressure conditions, while higher pressure experiments are also available [21]. The average velocity of airflow is 2.5 m/s. A summary of the experimental details is listed in Table 1.

In the experiment, a swirling flame is first established inside the combustion tube with the flame base located near the upper end of the center body. Flashback is then triggered by a step-increase of the fuel flow rate, resulting in a change in the global equivalence ratio from 0.5 to 0.63. During flashback, the flame propagates upstream entering the mixing tube. In the experiments [21], boundary layer flashback is observed, where the leading tip of the flame creeps down from the combustion chamber towards the swirler. Due to the swirl motion of the flow, the flashback also exhibits characteristics similar to a swirl flashback, with the flame root attaching only to the inner wall (center body), but not the outer wall. Since fuel mixing becomes less complete further upstream, the flame flashes back against an increasingly stratified mixture and decreasing effective flame speed, until the flashback is eventually arrested at a stream-wise location that is roughly half the mixing tube length.

The simulation domain is discretized using 20 million unstructured control volumes, dominated by hexahedral volumes, as shown in Figure 7. The cells cluster near the swirler and the mixing tube walls, with a minimum spacing of 0.11 mm. As no wall model is used in the current simulations, the boundary layer mesh has been refined extensively near the boundary layer to ensure that the first layer of control volume has a y+ value close to 1. A series of mesh convergence tests were carried out to ensure that the results do not depend on the mesh. It was found that additional refinement of the current mesh did not alter the flashback behavior, and produced insignificant quantitative differences.

A total of four cases were simulated to investigate stratified flame flashback, as listed in Table 2. Case A is designated as the baseline case. Case B and D are performed to compare the differences between the premixed and diffusion databases in capturing the flashback. As the wall temperatures were not explicitly measured in the experimental study, different wall temperatures are applied in cases A, B, and C to examine the effect of heat loss on flashback. The flashback is triggered using the same method as in the experiment; a stable flame is first established at a relatively low global equivalence ratio, which is followed by a step-increase in fuel flow rate (defined as t=0 ms) to trigger the flashback.

## 3. Results

### 3.1. Comparison between Simulations and Experiments

#### 3.1.1. Non-Reacting Results

Since flashback is an unsteady process that is challenging to investigate, the steady flow field is first examined without combustion and compared with non-reacting experiment data. Figure 8 shows the time-averaged axial velocity and the root-mean-square (RMS) axial velocity compared with experimental results at 42 mm upstream of the mixing tube exit (i.e., where it connects to the combustion tube). The mean axial velocity compares favorably with the experimental data. It is also seen that the profile is not symmetric with respect to the center of the annulus, indicating the role of the swirling vanes in altering the mean axial flow. The RMS profiles show that LES underpredicts turbulent fluctuations, which is consistent with the filtered representation and increased dissipation due to the sub-filter model. The asymmetry in the RMS profile indicates that the fluctuations near the inner wall are higher than that near the outer wall.

The statistical distribution of the equivalence ratio is measured within a sampling plane (27 mm × 13.5 mm) located 42 mm upstream of the mixing tube exit. Inside this sampling domain, two subdomains with radial locations r<6 mm and r>6 mm are defined. Both numerical and experimental results are shown in Figure 9. It can be seen that the spatial distribution of the equivalence ratio matches with experimental results in terms of both the range and the peak location of the probability density function of the equivalence ratio. It can also be seen that the fuel distribution is richer at the outer radius location of the mixing tube. This is primarily due to the radial placement of the fuel ports (explained in Section 2.3), combined with the swirling flow produced by the vanes, which generates higher velocity closer to the outer wall. The probability density of equivalence ratio matches well with experimental results in the inner radius subdomain, while minor discrepancies can be observed between the numerical and experimental results in the outer radius subdomain.

#### 3.1.2. Flashback Result from Baseline Case

In this section, the results of the Case A simulation are discussed. Figure 10 shows the instantaneous flame front location and associated fields after initiation of the flashback. There is considerable unmixedness close to the injection vanes, but the mixing increases towards the exit of the mixing tube. A high-temperature flame is established downstream with local temperatures indicating fully burnt condition. However, looking closer to the flame front, there exists considerable unmixedness, with a richer fuel–air mixture closer to the outer wall. Close to the inner wall, a region of negative axial velocity can be seen, which extends upstream of the flame location. This is consistent with findings in the experiment (discussed below). Other direct numerical simulation studies [23] of flashback in non-swirling flows show the generation of such negative velocity regions just ahead of the flame. Here, the negative velocity region is established at a far upstream location of the flame entering the mixing tube, indicating that the swirling vanes are the primary reason for creating this recirculation. In effect, the boundary layer near the inner wall is weakened by the orientation of the swirling vanes.

Figure 11 shows the key features during the upstream flame propagation. The flashback occurs along the center body dominated by the presence of the flame tongue (introduced in Section 1). The flame front can be further divided into the leading side and the trailing side based on the flame motion and the lowest axial location (i.e., the flame base [18]). The effect of wall temperature on this flame structure is discussed later in Section 3.2.2.

Figure 12 provides an overview of the velocity field and flame surface evolution from experiments and computations as the flame passes through the observation window (denoted by the vertical green line in the flame surface plots). Based on the bulk inflow velocity and the mixing tube length, the flow residence time in the mixing tube is roughly 65 ms (Figure 6 and Table 1). In Case A, the flame is arrested at about 75 mm and at t=630 ms. After accounting for the time taken by the increased fuel flow to reach the combustion chamber, the effective flashback velocity is roughly 0.133 m/s. The experimental result is only available for a single passage of flame surface through the measurement window of the laser sheet, located between 55 mm to 80 mm from the exit of the mixing tube. Hence, the flame arrest time and distance are unavailable, but it is known that the flame front at least gets to 77.5 mm upstream as measured from the end of the mixing tube. During the experimental measurement, the flame front takes about 5 ms to pass the experimental laser sheet (green vertical line), as shown in the top row of Figure 12. The region marked by the yellow circle in the experimental chemiluminescence images highlights the leading edge of the flame tongue entering the experimental laser sheet.

In the computations (third and fourth row of Figure 12), the first snapshot (t=560 ms) is selected from Case A when the flame surface (defined by T=1100 K) has reached a position comparable to the experiment image. The leading side of flame tongue enters laser sheets at 560 ms and flame base leaves the laser sheet at 565 ms, which is comparable to the experimental observation. The second row shows the experimental velocity contour measured from the laser sheet, and the numerical result obtained at the corresponding cutting plane. The PIV droplets evaporate behind the flame surface, and experimental velocity measurements are unavailable in this region. As the boiling point of the PIV droplets is known to be 490 K, the numerical flame surface shown by the white line in the fourth row is defined by the same iso-value. Relating the flame surface and velocity data, the experiments show that the flame becomes more convex with time (protruding into the unburnt mixture side), which is captured by the simulations, albeit with less volume occupied by the burnt region of the flame.

### 3.2. Parameter Studies

#### 3.2.1. Influence of Tabulation Approach

The results from simulations using premixed and diffusion databases are compared here (details on the tabulation are provided in Section 2). As indicated in Table 1, the premixed database does not predict flashback, while case B shows flame propagation upstream of the combustor. Here, the wall temperature is fixed at 1200 K for both cases in order to provide comparable tabulations. Figure 13 shows contours of key quantities obtained at identical simulation times. It is noted that the reaction progress variable distribution in the combustor section is different, which is also seen to some extent in the mixture fraction profiles. This difference is mainly due to the location of the flame front, which alters the local flow field, including the swirling flow and the mixing process. However, peak temperatures are nearly identical for both cases. Interestingly, there is very little difference in the mixture fraction fields close to the injection vanes. Note that even though the flow is subsonic, the density gradients imposed by the flame front do not appear to change the incoming flow upstream of the flame region.

To understand the differences in the flame behavior, a series of counterflow diffusion flames were simulated with a range of strain rates. In a previous study by Fiorina et al. [37], similar numerical experiments were performed to investigate the behavior of the premixed database under different combustion scenarios. Only the mixture fraction and progress variable are used to obtain the flame properties, and two sets of simulations utilizing the two tabulation methods were carried out. Figure 14 shows a comparison between premixed and diffusion databases. The peak reaction source obtained from the premixed database is larger than that from the diffusion database at a low strain rate of a=132 s${}^{-1}$, while an opposite trend is observed at higher strain rates of a=390 s${}^{-1}$ and a=580 s${}^{-1}$. Meanwhile, the peak reaction source obtained from the premixed database is at a *Z*-location that is richer than that from the diffusion database, which is more prominent at a lower strain rate. The same trend has been observed in [37]. Figure 15 shows the *Z*-space distributions of strain rate, reaction source, and reaction source probability distribution, obtained from the LES flashback simulations at the end time (1200 ms), where the data is sampled from all the control volumes in a section of the mixing tube where the flame front is present. There are two peaks of strain rate profiles in the result using the diffusion database, but only one in that using the premixed database. Regardless, the strain rates of both cases are lower than 132 s${}^{-1}$, which indicates that the flow dynamics here are similar to the set of low strain rate results (red symbols and lines) of the laminar flame in Figure 14. Further, from the reaction source probability distribution (third column of Figure 15), it can be observed that for the target configuration, the region where reaction source is important is on the fuel-lean side, concentrated around Z∼[0.035,0.04]. In this region, the tabulated reaction source obtained from the diffusion database is larger than that from the premixed database in the laminar flame calculation (red symbols and line in Figure 14), which is consistent with the reaction source distribution obtained from the flashback LES data (second column of Figure 15). This distribution of reaction source in mixture fraction space partially explains the reason that the diffusion database is able to capture the flashback process.

Finally, it is important to note that the ability of the diffusion database to predict flashback does not necessarily suggest that the flame structure is that of a diffusion flame. The LES reaction source distribution in Figure 15 only covers a *Z*-space region that is mostly on the fuel-lean side and within the rich flammability limit (see the vertical lines that indicate the distribution bounds of $Z_{min}$ and $Z_{max}$). Under such conditions, even when the premixed database predicts the right flame temperature and major combustion species, the reaction source prediction can be very different [37]. This inaccuracy is due to the importance of species diffusion across the isosurfaces of mixture fraction that increases with the species gradient and strain rate. However, in the cases considered here, the mixture is overall lean and experiences only low levels of strain. It is equally likely that the diffusion database overpredicts species diffusion in this regime. The premixed database based simulations could also be sensitive to the sub-grid PDF [48,49]. These results demonstrate that, unlike statistically stationary problems, the choice of the canonical flame configuration has a first-order impact on flashback prediction.

#### 3.2.2. Effect of Heat Loss on Flashback

To understand the role of wall temperature on the flashback process, three cases (A, B, and C in Table 2) are considered (Figure 16). The numerical flame front (iso-surface of 1100K) is plotted in the mixing tube section in order to show the differences in the upstream propagation. The flame in Case A enters the laser sheet (green vertical line) at 250 ms and is arrested at 630 ms, while the flame in Case C does not enter the observation window. It is seen that the flashback distance increases with increasing wall temperature, which will be further analyzed below. In particular, a wall temperature of 1500 K provides results that are consistent with the experimental data, which has already been discussed in Figure 12.

To better understand the impact of heat loss, planar data is extracted from the simulations as described in Figure 17. Two types of cutting planes are considered here: (a) the cylindrical surface near the center body wall, and (b) a series of azimuthal cutting planes. For the near-wall cylindrical surface, the LES instantaneous velocity field is extracted from a fixed wall distance of r=0.9 mm, which is chosen to be slightly larger than the quenching distance (∼0.1–0.5 mm). A similar strategy has been applied to investigate the flow–flame interactions by the previous experimental study of this configuration [18]. The extracted velocity contour is then unwrapped and projected onto a 2-d plane, with its horizontal coordinate being the azimuthal angle of the original cylindrical coordinate system and its vertical axis being the original *z*-axis, as shown by the upper row of Figure 17. This procedure allows the identification of key features of the flow. In particular, the flame tongue can be identified as the section of the flame surface that is preceded by a prominent negative axial velocity zone, which is always placed in the plot center (defined as θ=0) for each case.

An overview of the flashback physics is first presented for the three cases of different wall temperatures (Case A, B, and C) in Figure 18, Figure 19 and Figure 20. For all cases, the time instance where the snapshots are recorded is t=330 ms after flashback is triggered. At this time instance, flashback has not been arrested for any of the three cases yet, and as the fuel injection and mixing processes are comparable among all cases, the differences observed are mostly due to heat loss. From the unwrapped cylindrical surface plot, a distinguishable flame tongue can be observed for all three cases, featured by a right-tilted (swirling flow moves from left-bottom to right-top) flame surface, suggesting that flashback is marked by the upper-right edge of the flame tongue. From the axial-radial cutting plane plots, it can be seen that the slices corresponding to the flame tongue feature a low or even negative axial velocity region with an appreciable size that spans over a range of azimuthal angles (∼90${}^{\circ }$–120${}^{\circ }$).

For all cases, traveling from the leading edge of the flame tongue to the flame base (right to left), the flame surface (black solid lines) is initially broadly placed in the azimuthal plane, occupying a larger fraction of the flow field, but slowly becomes narrow and elongated, and is eventually completely washed down behind the negative axial velocity zone (white solid lines) when reaching the flame base. This reflects the critical role of the negative axial velocity region during flashback: near the leading edge (right side) of the flame tongue, the negative velocity zone is protected from direct contact with the incoming flow (from bottom-left to upper-right), where the flow is appreciably decelerated. The bottom-left edge of the negative axial velocity zone reduces its flashback propensity by becoming elongated such that the right edge of the negative velocity zone is able to maximize propagation in the radial direction. This behavior is qualitatively the same among for different wall temperatures and is also similar to experimental observations for this configuration [17,18]. The slices obtained at other angular positions feature a less prominent low/negative axial velocity region. However, the flame surface there also exhibits a convex shape (e.g., the second slice from the right in the bottom plot of Figure 18 similar to the flame tongue, which indicates that the local flame surface in those slices also propagates upstream. These small-scale structures are the above-mentioned flame bulges, which provide only temporary propagation upstream, but do not cause sustained flashback and are eventually pushed downstream by the oncoming flow.

Comparing the results of different wall temperatures, the size of the flame tongue is similar for all cases in terms of the spread in the angular direction. However, the size of the negative axial velocity region is considerably smaller with a lower temperature, which is most observable in the azimuthal plane, especially for Case C. From the axial velocity contours, it can be observed that the post flame velocity is also lower for a lower wall temperature, even when the flame position is similar, which suggests that the blockage effect cased by the post-flame thermal expansion plays a role in the formation of the negative velocity region.

Case B shows some additional features not prominently found in Case A and C. In particular, two different flame tongues are observed, which seems to indicate that this feature alone is not the dominant cause of flashback. There are two possible explanations for this secondary feature: there is a competing effect between stratification and heat loss, which will be further explored below, or that this snapshot captures a transition period from one flame tongue to another, which is less likely but has been reported in the experimental study as well [17].

To further investigate the mechanism behind the swirl BL flashback, the spatial distributions of other flow field properties, including mixture fraction, reaction source, and enthalpy defect, are plotted in Figure 21, Figure 22 and Figure 23. Note that the contours here cover only the inner half of the annulus in the radial direction (r<7.3 mm). In this inner radius region, the mixture is fuel-lean (Section 3.1.1), until reaching an upstream location of z<−60 mm, where the local fuel-rich pockets can exceed the stoichiometric fuel/air ratio ($Z_{st}=0.055$). It can be further observed that the fuel stratification here exhibits large-scale fluctuations in the azimuthal direction with a scale ($\theta ∼O(90^{{}^{\circ}})$) that is of the same order of magnitude as the span of the flame tongue. Especially for Case A (top row of Figure 21) and C (top row of Figure 23), the contour slices at angular positions corresponding to the flame tongue exhibit an overall higher mixture fraction value compared to other azimuthal slices. Consequently, the axial velocity contours in previous Figure 18 and Figure 20 exhibit a higher post-flame axial velocity in those same regions. This observation is consistent with the view that the formation of the negative axial velocity zone is related to the thermal expansion effects, which is here found to be further related to the local fuel/air ratio.

Case B results (Figure 22) show a slightly different behavior, with the main flame tongue exhibiting overall lean equivalence ratio, but the second flame tongue is located in a region with richer fuel/air mixtures. The enthalpy defect contours of Case B (last row of Figure 22) show that the secondary flame tongue features a radially larger region of large heat loss, especially near the lowest tip of the flame surface, whereas the main flame tongue features a comparatively insignificant heat loss region. This atypical behavior in the secondary flame tongue region is due to higher wall heat loss in spite of the richer fuel/air mixture. As a result, the negative velocity is lower and is also radially smaller compared to the main flame tongue. However, the difference between case B and the other two cases shows that the flashback physics is sensitive to the joint effects of fuel stratification and wall heat loss.

Focusing on the specific effect of wall temperature, it can be observed that for Case A, comparing the mid and bottom row of Figure 21, the heat loss region rarely interferes with the region where the flame surface is interacting with the negative axial velocity region. As a result, the flame surface there is not experiencing any significant flame quenching. In Case B, the heat loss effects start becoming prominent enough to quench some of the flame surface near the negative axial velocity region, especially for the secondary flame tongue as already discussed. With Case C, it is seen that the reaction source terms are considerably lower in the region close to the flame tip, indicating significant flame quenching.

## 4. Conclusions

In this work, transient boundary layer flashback in lean stratified methane–air swirl flames was studied computationally using non-adiabatic tabulated models in the LES framework. The effect of near-wall flame quenching through conductive heat transfer was incorporated in the tabulation procedure. Look-up tables with premixed and diffusion flames were adopted here in order to explore the role of the canonical flame configuration on flashback prediction. Simulations with the different flame tabulations and wall temperatures were conducted.

The flame tabulation studies indicate that the diffusion table accurately captures the flame development and flashback while the premixed table fails to capture upstream flame propagation. Overall, the flow configuration operates in the low strain rate fuel-lean zone. Under these conditions, the diffusion table generates a higher source term for the progress variable compared to the premixed database. Moreover, the simulation with the diffusion database captures some of the key characteristics observed in experiments, including the flame tongue and flame front wrinkling. It is also found that the leading edge of the flame tongue plays a critical role in the flashback processes. In particular, the generation of the negative velocity region upstream of the propagating flame front aids in flashback.

Comparing the simulations with different wall temperatures, it is seen that heat loss increases the flame quenching process and arrests upstream propagation. The negative velocity region occupies a smaller volume with increased heat loss, which limits upstream propagation. Interestingly, multiple flame tongues are found for particular wall temperatures. However, the secondary flame tongues do not generate negative axial velocity regions, mainly due to the competing effects between fuel stratification and heat loss. The key physical features in the simulations reproduce experimental observations.

## Figures and Tables

**Figure 1 entropy-23-00567-f001:**
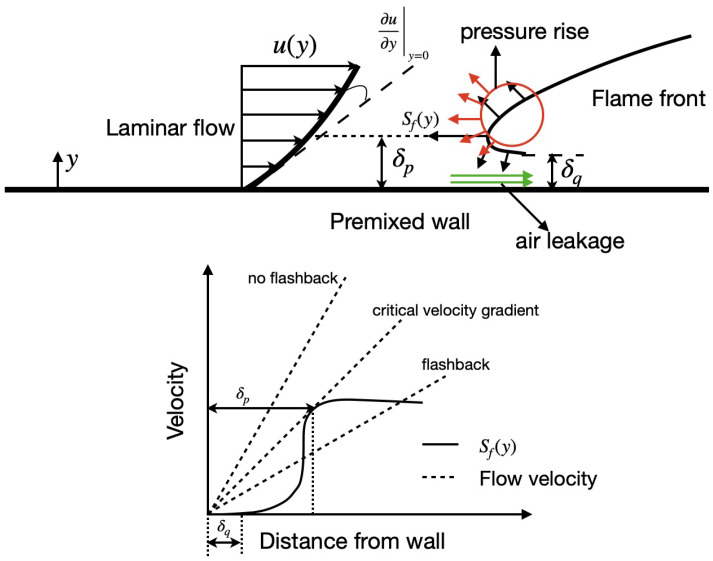
(**Top**) A schematic of upstream flame propagation near a wall, identifying the key parameters. (**Bottom**) Critical gradient for flashback, reproduced from [6]. Here, $\delta_{p}$ is the distance from the wall where the local flame velocity matches the fluid velocity, and $\delta_{q}$ is the flame quenching distance.

**Figure 2 entropy-23-00567-f002:**
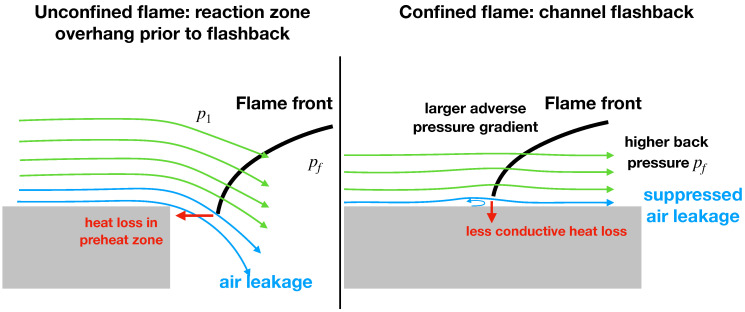
A schematic of flame propagation in (**left**) unconfined and (**right**) confined configurations. Reproduced from [2].

**Figure 3 entropy-23-00567-f003:**
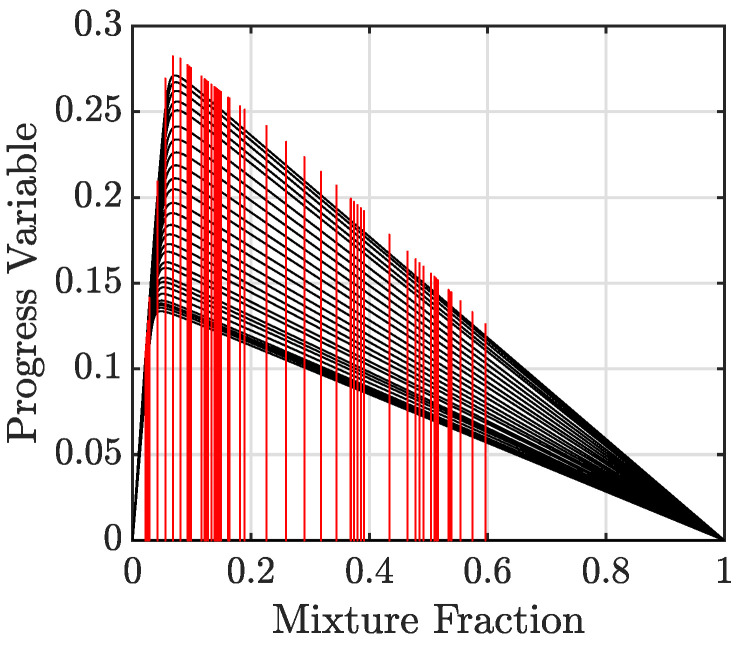
Solutions of premixed flamelets (—) and diffusion flamelets (—) plotted in the tabulation phase space.

**Figure 4 entropy-23-00567-f004:**
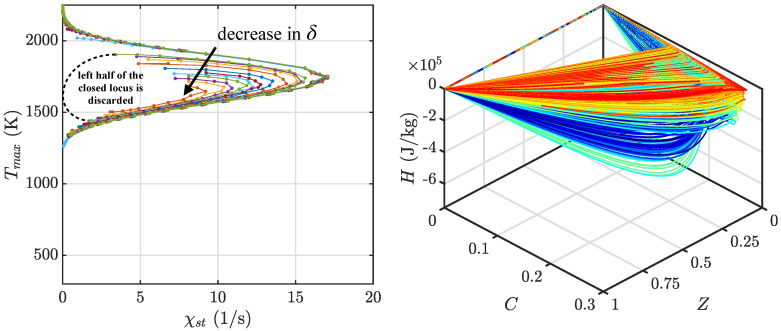
Sample of the tabulation using counterflow diffusion flamelets with additional energy sink term $\dot{q}$ in Equation (Equation 1) parameterized by δ. (**Left**) Loci of maximum flamelet temperature as a function of scalar dissipation rate at stoichiometric conditions, plotted by varying δ. (**Right**) Flamelet solution in the tabulation phase space, with line colors indicating a decrease in δ while going from red to blue.

**Figure 5 entropy-23-00567-f005:**
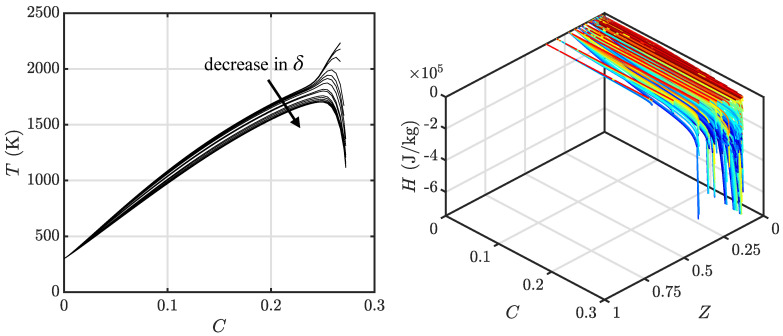
Sample of the tabulation using 1d freely propagating flames with additional energy sink term $\dot{q}$. (**Left**) Flamelet temperature profiles obtained at stoichiometric equivalence ratio. (**Right**) Solutions plotted in tabulation phase-space for decreasing values of δ while going from red to blue.

**Figure 6 entropy-23-00567-f006:**
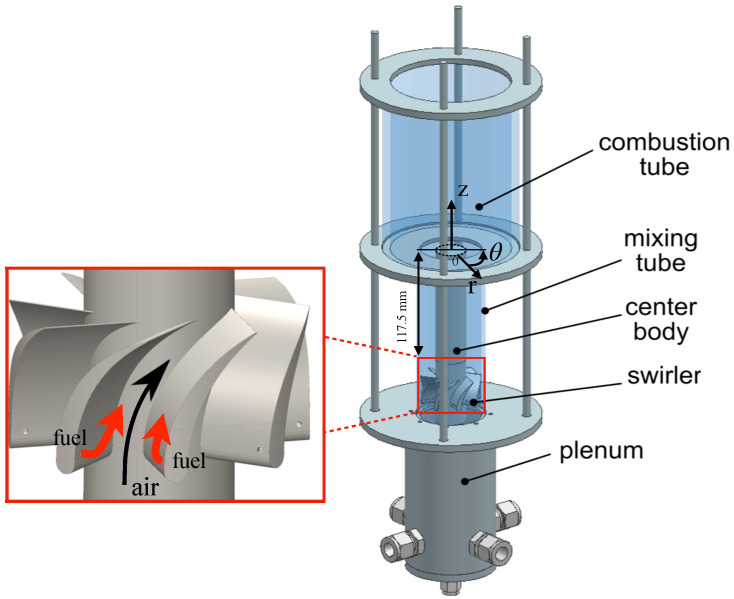
Schematic of the experimental swirl combustor, reproduced from [21].

**Figure 7 entropy-23-00567-f007:**
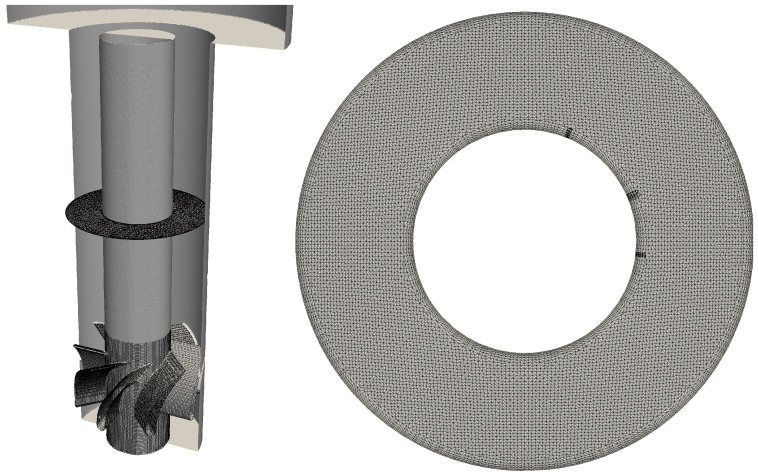
Samples of the computational mesh used in this study.

**Figure 8 entropy-23-00567-f008:**
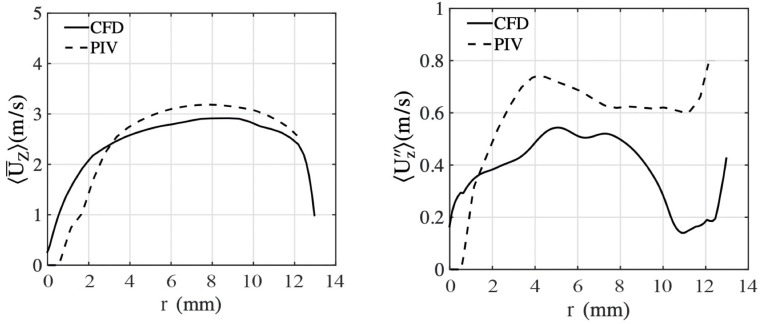
Comparison of mean and fluctuation of axial velocity between simulations (lines) and experimental measurements (dotted lines). Note that r=13.6 mm denotes the outer wall, while r=0 mm denotes the inner wall.

**Figure 9 entropy-23-00567-f009:**
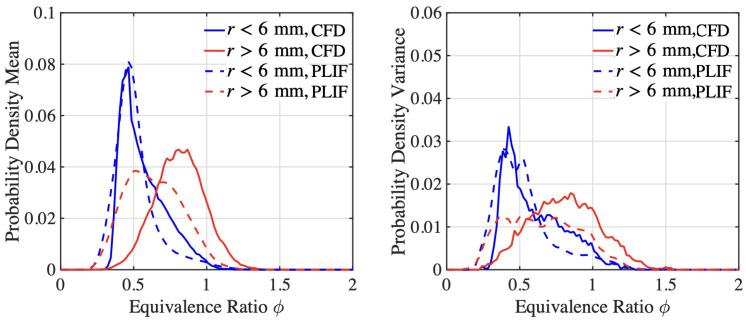
Time-averaged (**left**) equivalence ratio distribution and (**right**) RMS fluctuations.

**Figure 10 entropy-23-00567-f010:**
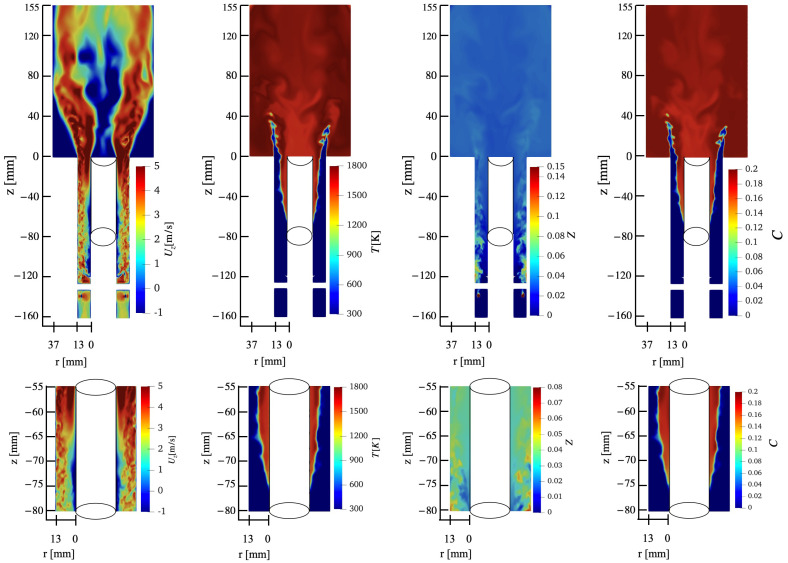
(**Top**) Instantaneous contours from an azimuthal plane for CASE A at t=478 ms, and (**Bottom**) zoomed in view of the same data. The different columns show velocity, temperature, mixture fraction, and progress variable, respectively.

**Figure 11 entropy-23-00567-f011:**
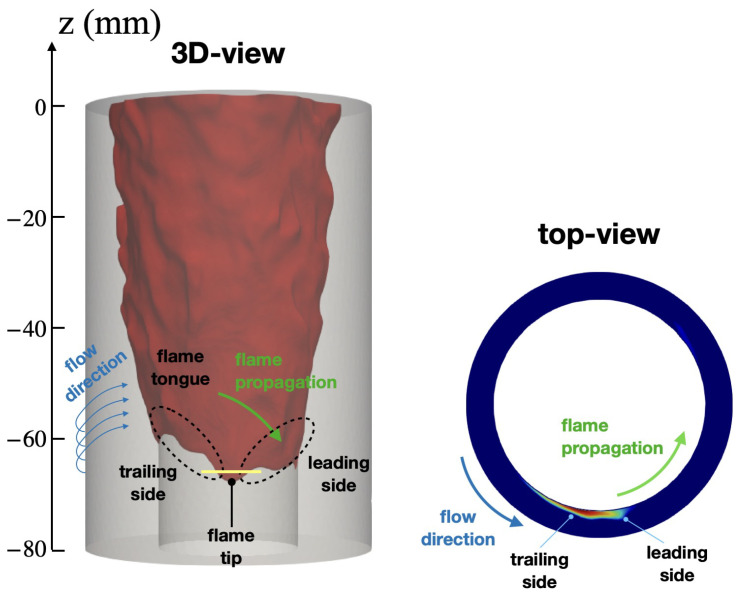
Instantaneous flame surface showing the flame tongue during flashback obtained from CASE A. (**Left**) 3D view and (**Right**) cutting plane of z=−68 mm at t=554 ms. Yellow line in 3D view indicates the axial location of top view cutting plane.

**Figure 12 entropy-23-00567-f012:**
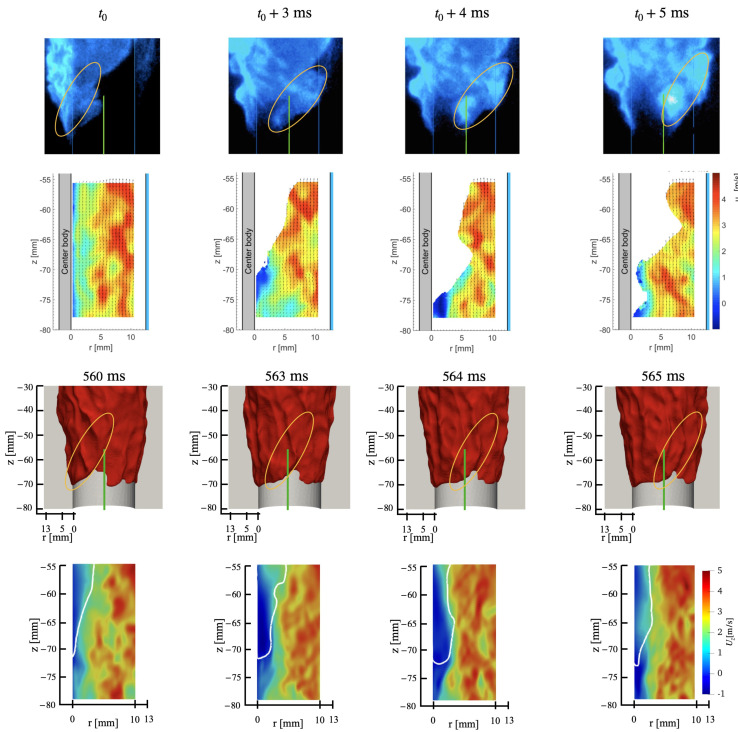
Comparison of flame evolution and the axial velocity fields from experiment and case A simulation. The four rows (top to bottom) show experimental chemiluminescence, experimental axial velocity, numerical flame surface defined by 1100 K isosurface of temperature, and numerical axial velocity, respectively. Yellow circles mark the leading edge of the flame tongue.

**Figure 13 entropy-23-00567-f013:**
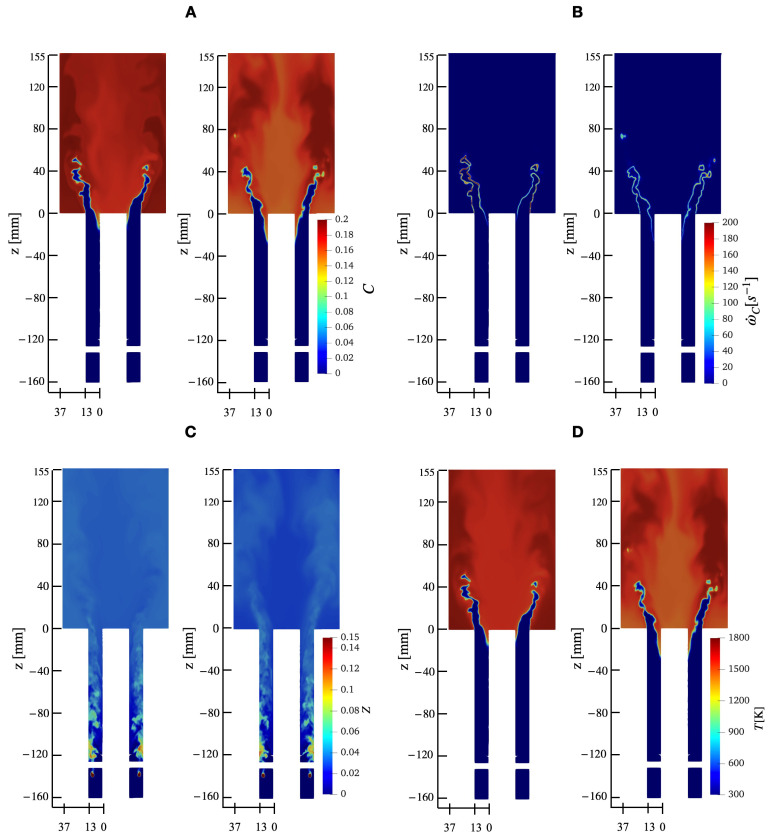
Comparison of instantaneous fields obtained using premixed database and non-premixed databases, both sampled at 1200 ms. (**A**) progress variable, (**B**) source term of progress variable, (**C**) mixture fraction, and (**D**) temperature. For each quantity, the left contour plot is obtained from premixed database while the right plot is obtained using the diffusion database.

**Figure 14 entropy-23-00567-f014:**
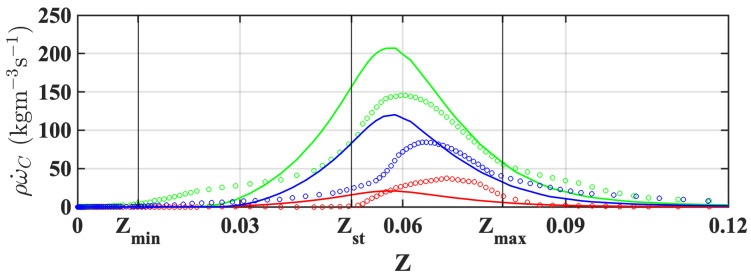
Comparison of progress variable source term obtained from one-dimensional counterflow diffusion flames using premixed and diffusion databases and for different strain rates *a*. (Lines) diffusion database; (Symbols) premixed database. — and ○: a=580 s${}^{-1}$; — and ○: a=390 s${}^{-1}$; — and ○: a=132 s${}^{-1}$.

**Figure 15 entropy-23-00567-f015:**
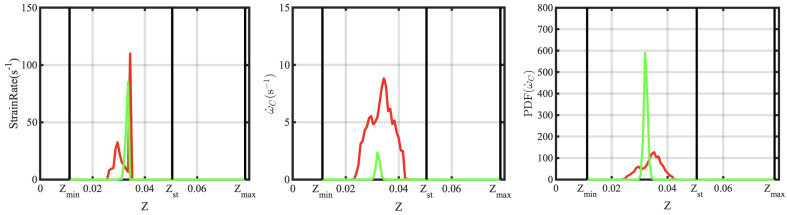
*Z*-space variations of strain rate, reaction source, and reaction source probability distribution at the end state (1200 ms), sampled from all the control volumes in the mixing tube section from 0 mm to −80 mm of mixing tube exit using LES simulations with premixed database (—) and diffusion database (—). $Z_{min}$ and $Z_{max}$ mark the lower and upper bounds of *Z*-space covered by the sampled LES data points, respectively.

**Figure 16 entropy-23-00567-f016:**
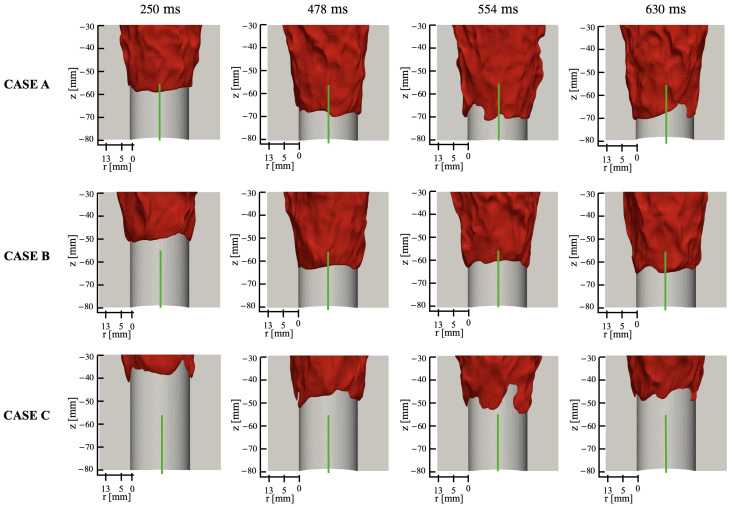
Time series of flame surface (T=1100 K) from LES calculations obtained using different wall temperature parameters. The three rows correspond to the three cases, while the columns show the surface at different times (marked at the top).

**Figure 17 entropy-23-00567-f017:**
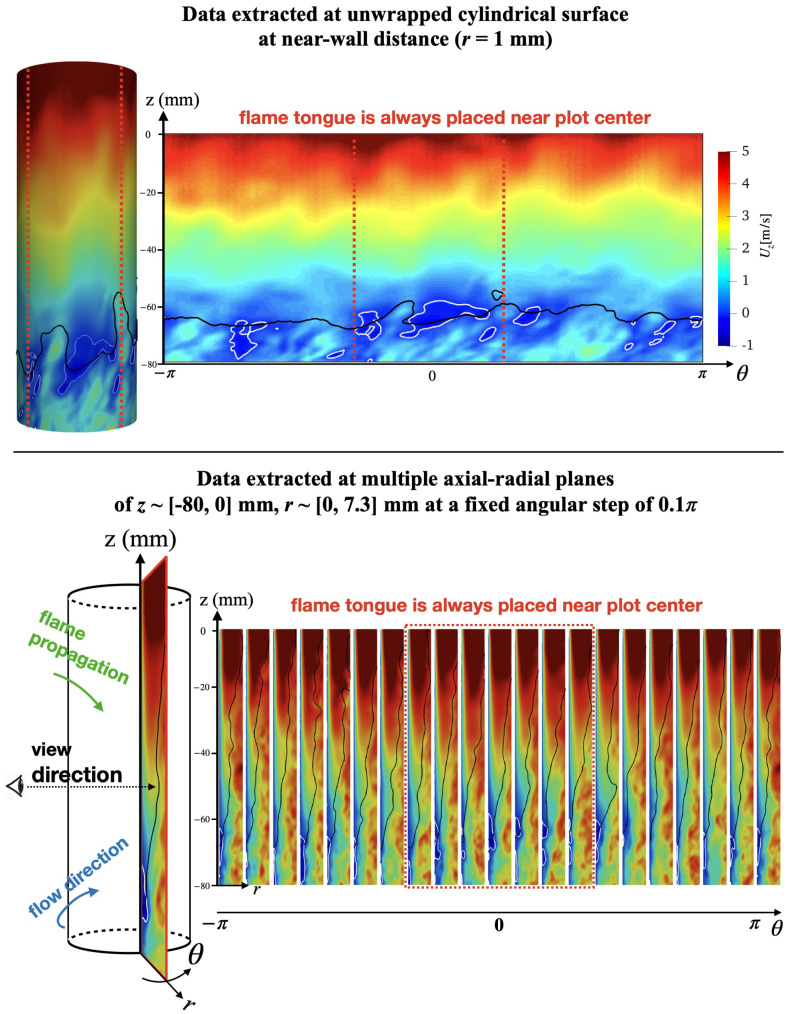
Sample image of the unwrapped flow field (**upper** plot) and azimuthal slice of the velocity (**bottom** plot). Black line is the isoline of T=1100 K and denotes flame front. White line is the $U_{z}=0$ m/s isoline and indicates the boundary layer separation zone.

**Figure 18 entropy-23-00567-f018:**
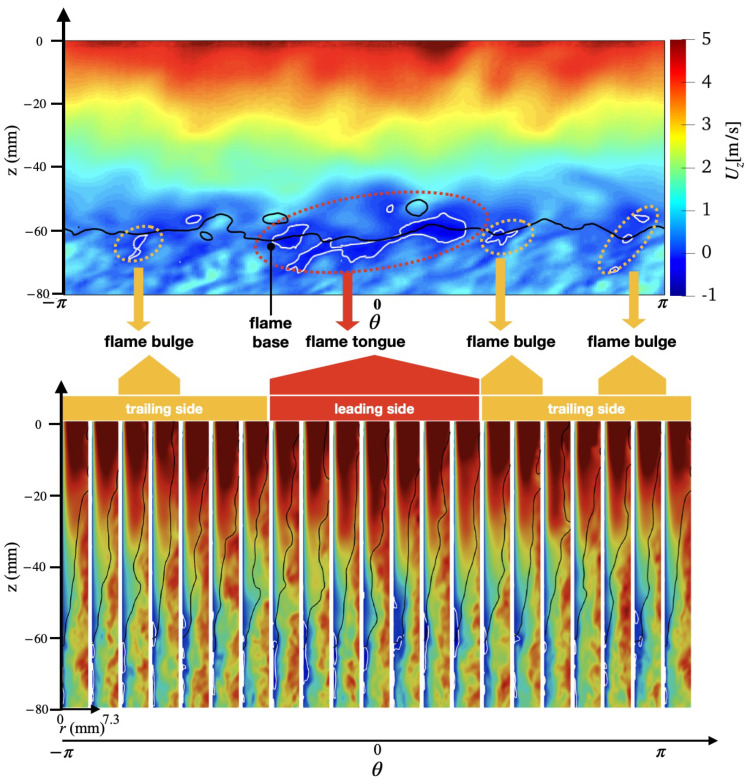
(**Top**) Unwrapped flow field and (**Bottom**) azimuthal slices for case A at t=330 ms.

**Figure 19 entropy-23-00567-f019:**
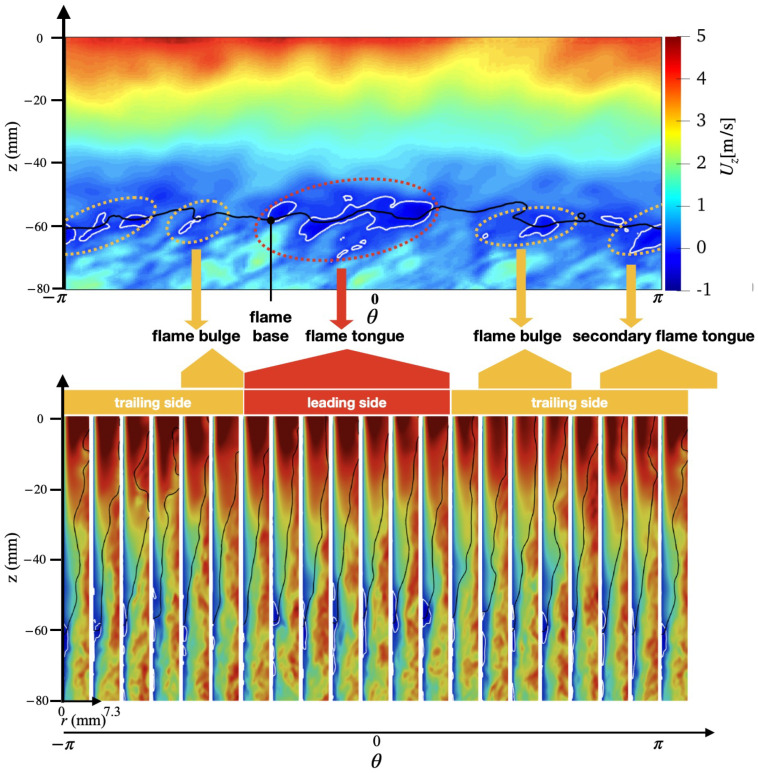
(**Top**) Unwrapped flow field and (**Bottom**) azimuthal slices for case B at t=330 ms.

**Figure 20 entropy-23-00567-f020:**
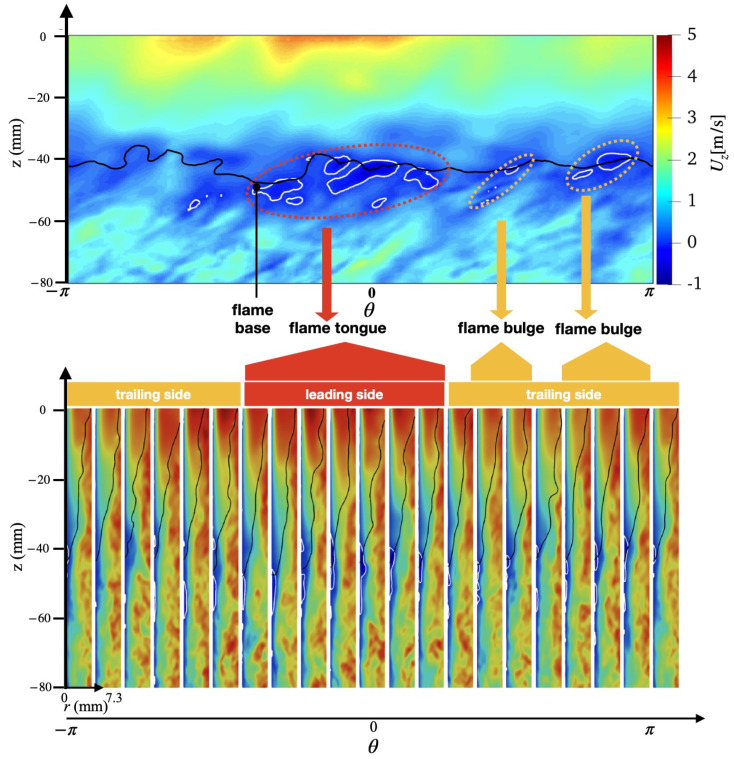
(**Top**) Unwrapped flow field and (**Bottom**) azimuthal slices for case C at t=330 ms.

**Figure 21 entropy-23-00567-f021:**
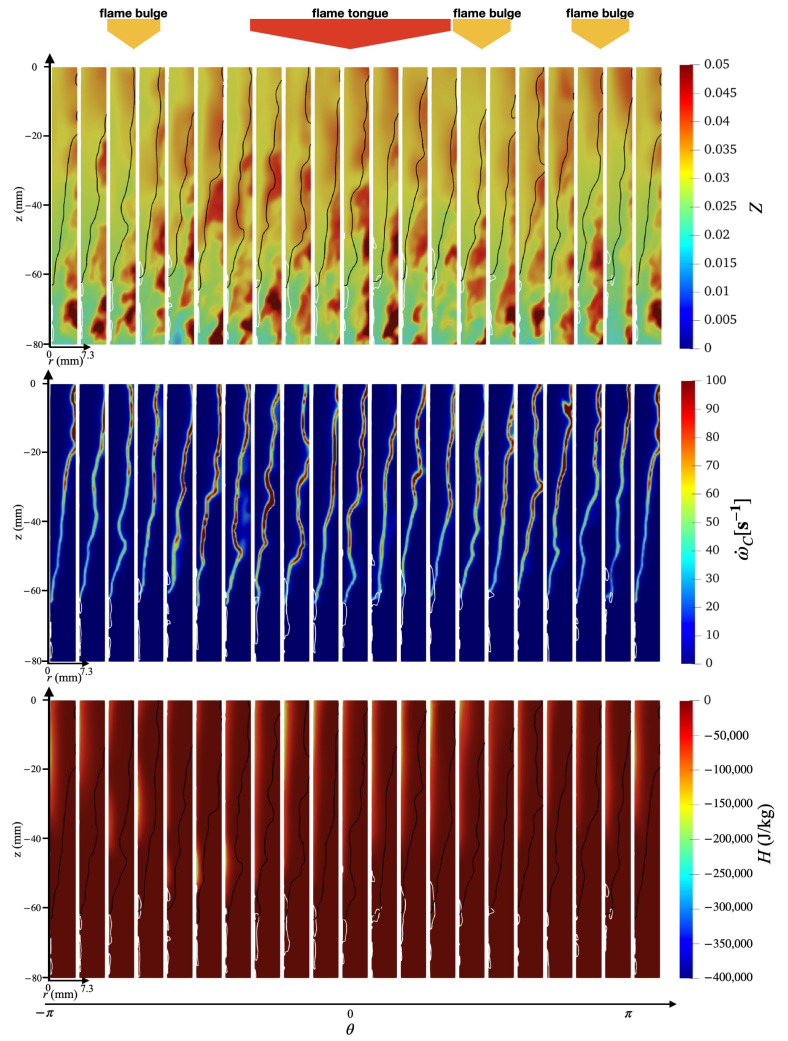
Azimuthal slices of instantaneous snapshot at t=330 ms for case A obtained using non-premixed database. (**Top**) Mixture fraction, (**middle**) reaction source term, and (**bottom**) enthalpy defect.

**Figure 22 entropy-23-00567-f022:**
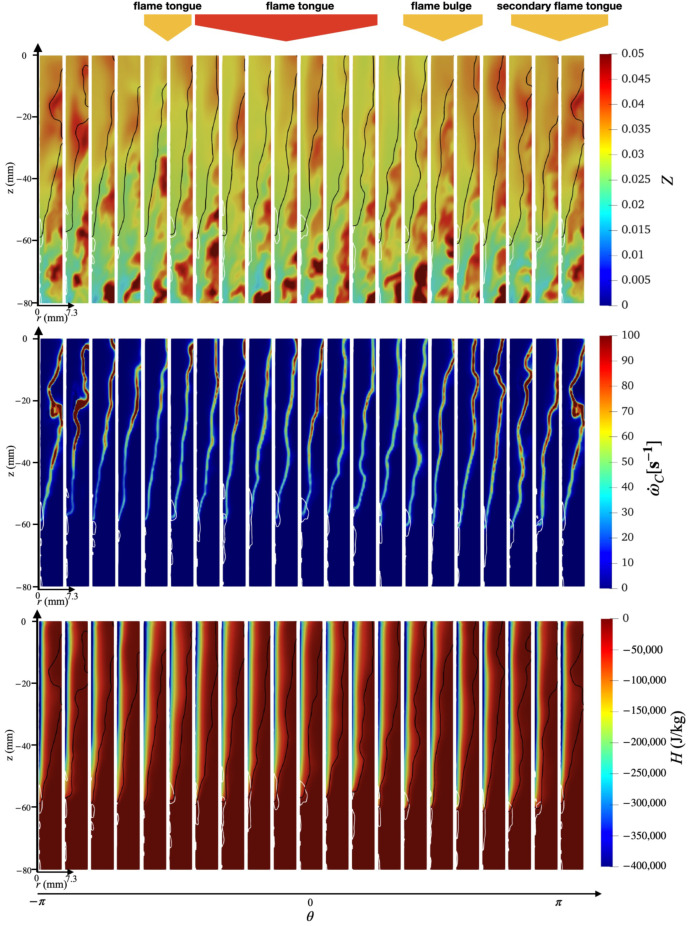
Azimuthal slices of instantaneous snapshot at t=330 ms for case B obtained using non-premixed database. (**Top**) Mixture fraction, (**middle**) reaction source term, and (**bottom**) enthalpy defect.

**Figure 23 entropy-23-00567-f023:**
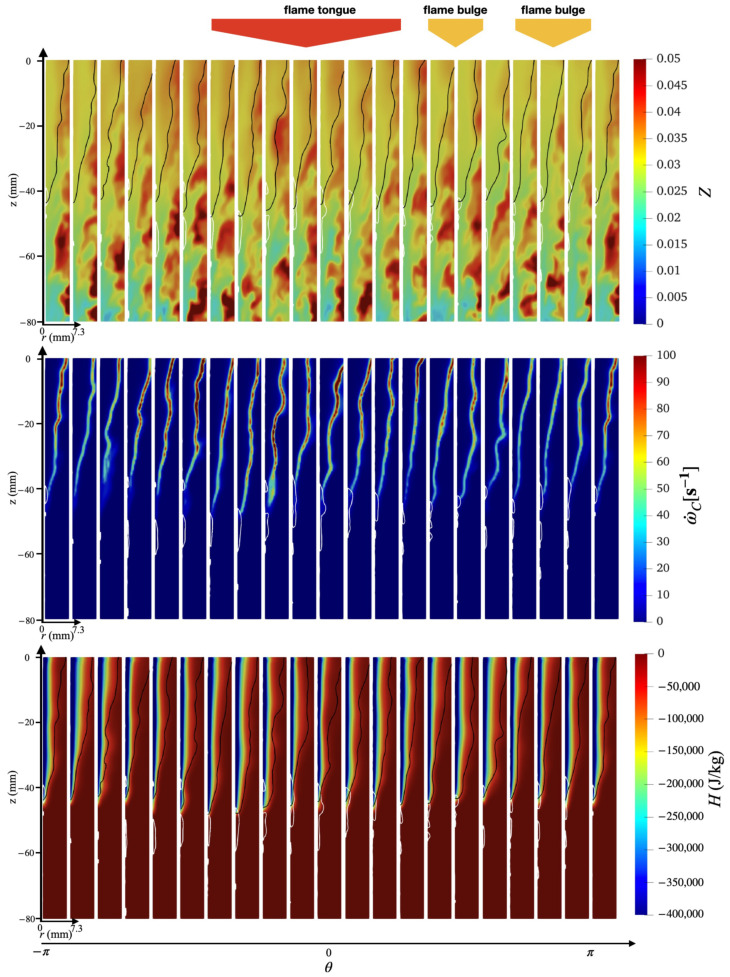
Azimuthal slices of instantaneous snapshot at t=330 ms for case C obtained using non-premixed database. (**Top**) Mixture fraction, (**middle**) reaction source term, and (**bottom**) enthalpy defect.

**Table 1 entropy-23-00567-t001:** Experimental operating conditions.

Properties	Value
Fuel	CH4
Inflow bulk velocity	2.5 m/s
Stable equiv. ratio	0.5
Flashback equiv. ratio	0.63
Inflow temperature	300 K
Pressure	1 bar
Reynolds numbers	6100

**Table 2 entropy-23-00567-t002:** Summary of test cases and flashback results.

Case ID	$T_{w}$	Tabulation Model	Flashback Occurrence	Flashback Distance ^a^	Arrested Time ^b^
A	1500 K	diffusion database	yes	75 mm	630 ms
B	1200 K	diffusion database	yes	65 mm	590 ms
C	1000 K	diffusion database	yes	52 mm	478 ms
D	1200 K	premixed database	no	-	-

${}^{a}$ The flashback distance is measured from the tip of the flame surface (defined by the temperature iso-value of T=490 K) to the mixing tube exit after the upstream propagation has stopped. In the experimental study, the flashback distance is estimated to be 77.5 mm. ${}^{b}$ For the numerical results, t=0 ms is defined as the time when flashback is triggered by a step-increase in fuel flow rate.

## Data Availability

Not applicable.

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
