# Peer review of "Computational Modeling of Boundary Layer Flashback in a Swirling Stratified Flame Using a LES-Based Non-Adiabatic Tabulated Chemistry Approach"

_entropy, 2021, doi:10.3390/e23050567_

Round 1

Reviewer 1 Report

The article concerns very interesting analyzes related to the modeling of flows in the combustion process. The main goal of this work is to develop a comprehensive simulation approach to the boundary layer flashback model that takes into account fuel and air stratification and wall heat losses. A structure based on the simulation of large eddies (LES) with a tabular combustion model is used. The obtained results were experimentally verified using PIV. For this reason, the article is very interesting and important. The article is written at a very good substantive level. Nevertheless, in my opinion, it requires editing in line with the journal's standards. The authors write for modeling the flow of burners or combustion chambers of gas turbines, and from my knowledge the pressure in these devices significantly exceeds the pressure of 1 bar (in the article Table 1) - usually it is several or even several dozen bar. Please clarify whether the models proposed in the article will reflect combustion processes at increased pressure. In conclusion, I consider the assessed article to be made at the appropriate substantive level for publication in the Entropy journal and recommend it for publication, of course after appropriate editorial formatting. 

Author Response

Dear Reviewer 1,

Thank you for your time and comment. Please find our detailed response attached.

Regards,
Xudong

Reviewer 2 Report

The manuscript is to develop a comprehensive simulation approach to model boundary layer flashback, accounting for fuel-air stratification and wall heat loss. The authors have done a comprehensive study and the paper is very well structured. The methodology is described in detail and the analysis of the results is illustrative and traceable. The obtained results are also validated against experimental data to confirm the effectiveness of the proposed approach. 

Overall, the paper is very well written and could be fruitful for researchers in the field. In my view, the manuscript could be considered for publication after addressing the following minor room by the authors: 

1- The literature review is comprehensive and detailed. A more highlighted gap analysis to justify the necessity of doing this piece of research would be welcomed (e.g. to briefly describe the relative merits of the proposal to other publicly available approaches; the pros and cons, limitations and assumptions, etc.)

Author Response

Dear Reviewer 2,

Thank you for your time and comment. Please find our detailed response attached.

Regards,
Xudong
